# Heavily-Doped Bulk Silicon Sidewall Electrodes Embedded between Free-Hanging Microfluidic Channels by Modified Surface Channel Technology

**DOI:** 10.3390/mi11060561

**Published:** 2020-05-31

**Authors:** Yiyuan Zhao, Henk-Willem Veltkamp, Thomas V. P. Schut, Remco G. P. Sanders, Bogdan Breazu, Jarno Groenesteijn, Meint J. de Boer, Remco J. Wiegerink, Joost C. Lötters

**Affiliations:** 1MESA+ Institute for Nanotechnology, University of Twente, 7522 NB Enschede, The Netherlands; h.veltkamp@utwente.nl (H.-W.V.); t.v.p.schut@utwente.nl (T.V.P.S.); r.g.p.sanders@utwente.nl (R.G.P.S.); b.breazu@student.utwente.nl (B.B.); m.j.deboer@utwente.nl (M.J.d.B.); r.j.wiegerink@utwente.nl (R.J.W.); j.c.lotters@utwente.nl (J.C.L.); 2Bronkhorst High-Tech BV, 7261 AK Ruurlo, The Netherlands; j.groenesteijn@bronkhorst.com

**Keywords:** Surface Channel Technology (SCT), free-hanging microfluidic channels, silicon sidewall electrode, heavily-doped bulk silicon microheaters

## Abstract

Surface Channel Technology is known as the fabrication platform to make free-hanging microchannels for various microfluidic sensors and actuators. In this technology, thin film metal electrodes, such as platinum or gold, are often used for electrical sensing and actuation purposes. As a result that they are located at the top surface of the microfluidic channels, only topside sensing and actuation is possible. Moreover, in microreactor applications, high temperature degradation of thin film metal layers limits their performance as robust microheaters. In this paper, we report on an innovative idea to make microfluidic devices with integrated silicon sidewall electrodes, and we demonstrate their use as microheaters. This is achieved by modifying the original Surface Channel Technology with optimized mask designs. The modified technology allows to embed heavily-doped bulk silicon electrodes in between the sidewalls of two adjacent free-hanging microfluidic channels. The bulk silicon electrodes have the same electrical properties as the extrinsic silicon substrate. Their cross-sectional geometry and overall dimensions can be designed by optimizing the mask design, hence the resulting resistance of each silicon electrode can be customized. Furthermore, each silicon electrode can be electrically insulated from the silicon substrate. They can be designed with large cross-sectional areas and allow for high power dissipation when used as microheater. A demonstrator device is presented which reached 119.4
${}^{\circ }C$ at a power of 206.9
mW, limited by thermal conduction through the surrounding air. Other potential applications are sensors using the silicon sidewall electrodes as resistive or capacitive readout.

## 1. Introduction

### 1.1. Surface Channel Technology

Silicon-based microfluidic channels (i.e., microchannels) have been developed for various applications such as mass flow sensors [1], biosensors [2], and chemical reactors [3]. Depending on the silicon micromachining technology, microchannels with different geometries and dimensions can be made and different types of electrodes can be integrated. For example, microchannels with large cross-sectional areas can be realized by bonding two wafers with cavities together and thin film metal or poly-crystalline silicon electrodes can be integrated [4,5,6,7,8,9,10]. Microchannels with small cross-sectional areas can be made by selective removal of sacrificial layers to form the channel cavity and thin film electrodes can be integrated as sensing elements [11,12,13]. Free-hanging microchannels with intermediate cross-sectional areas can be made by bulk-micromachining of the silicon substrate, which allows integration of both thin film and bulk silicon electrodes [14,15,16,17,18,19].

It is desirable to develop a microfluidic platform [20] which allows various flow sensing functionalities to be realized using the same micromachining technology in a single wafer. Surface Channel Technology (SCT) has been developed for various flow sensing and flow reactor applications, such as micro-Coriolis flow sensors [18,19], fluid parameter sensors [21], control valves [22], pressure sensors [23], thermal flow sensors [24], and micro-gas-burners [25]. Surface channel technology allows the fabrication of free-hanging microfluidic channels with hydraulic diameters ranging from approximately 20 μm to 100 μm [16]. These microchannels have very thin channel walls in the range of 1 μm to 1.5 μm thickness and are made using low pressure chemical vapor deposition (LPCVD) of silicon-rich silicon nitride (SiRN). The microchannels can be completely released from the bulk silicon. Furthermore, using refilled trenches it was shown that silicon sidewall microheaters can be integrated in a microreactor with the Trench-Assisted Surface Channel Technology (TASCT) [26]. In this paper, we report an extension to the standard SCT process to integrate silicon sidewall electrodes between adjacent free-hanging microchannels in a silicon wafer.

### 1.2. Demands for Sidewall Microelectrodes

The resulting channels made with the original SCT process, as shown in Figure 1a, are free-hanging and allow for mechanical movement for sensing and actuation purposes [16]. All electrical functionalities are realized by thin film metal electrodes on top of the microchannels. For example, resistive strain gauges [23], temperature sensors [24], thin film microheaters [25], and capacitive readouts [26,27] have been implemented using this approach. In these cases, sensing and actuation is only possible from the topside.

Having electrodes only on top of the microchannels limits the device design freedom and several attempts were made to embed electrodes in the sidewalls of the channels [26,27]. For example, in-line relative permittivity sensors were realized with the SCT process in an SOI wafer (Figure 1b) by isolating two silicon electrodes at the sidewalls of the microchannels [27]. Accurate capacitance readout was demonstrated, suggesting the viability of silicon electrodes for capacitive readout [27]. However, an SOI wafer was needed to electrically isolate the device layer silicon electrodes from the bulk handle layer.

The concept of silicon sidewall electrodes has another potential application as robust and reliable sidewall microheaters. For the micro-gas-burner presented in [25], 200 nm thick platinum resistors were patterned on the top surface of the microchannels and functioned as topside microheaters and temperature sensors. Although the high melting temperature of platinum ( 1769 ${}^{\circ }C$) makes it a promising candidate to be used as heater material, thin film platinum degrades morphologically and electrically at temperatures above 600 ${}^{\circ }C$ and therefore it cannot be used as a reliable microheater for high temperature flow reactor applications [25,28]. Large surface-to-volume-ratios in the platinum thin film electrodes drive fast diffusion and agglomeration at elevated temperatures [29]. The degraded platinum thin films become discontinuous and form separated islands which gives unstable electrical resistances and limits the use of thin film platinum at high temperature applications [30]. Moreover, at elevated temperatures, thermal stresses cause the platinum thin film to have hillock formation, delamination, buckling, and cracking problems [31]. This is due to the large differences in the coefficients of thermal expansion between the platinum and the silicon or the ceramic substrate [32]. An adhesion layer such as titanium or tantalum [33] is needed [34]. Many efforts have been made to enhance the performance of thin film platinum microheaters at high operation temperatures by optimizing the film thickness, encapsulation, and performing post-deposition high temperature annealing [28,30,35,36,37]. Compared to the thin film platinum microheaters, the advantages of using bulk silicon as microheaters will be discussed in details in Section 3.1.

It was previously reported that silicon microheaters can be embedded in the channel sidewalls by the TASCT process (Figure 1c) and temperatures up to 400 ${}^{\circ }C$ were reached by Joule heating, limited by the absence of flexure structures to allow for thermal expansion [26]. In order to integrate silicon sidewall electrodes, the currently existing methods either use SOI wafers [18,27] and/or refilled trenches [26], which add more complexity to the fabrication process.

Here we propose to modify the original SCT by introducing heavily-doped bulk silicon electrodes between the sidewalls of adjacent free-hanging microchannels, i.e., silicon sidewall electrodes. A cross-sectional view is schematically illustrated in Figure 1d. The proposed modified SCT fabrication technology is completely based on the core concept of the original SCT in a silicon wafer. Therefore, the modified SCT technology still allows all earlier designs [18,19,21,22,23,24,25,38] to be fabricated using the same SCT platform.

### 1.3. Outline

In the following sections, firstly, in Section 2.1, the modified SCT fabrication process will be explained. Next, in Section 2.2, we will present test structures that were designed, fabricated, and analyzed to optimize the relevant design parameters. Furthermore, in Section 3, we will present a demonstrator device that was designed to use the silicon sidewall electrodes as microheaters. Lastly, in Section 4, the development of the modified technology is summarized and an outlook on further research is given.

## 2. Modified Surface Channel Technology

### 2.1. Fabrication Process

The modified SCT fabrication process is schematically illustrated in Figure 2 showing the cross-sectional views along the channel length (cross-section 1) and perpendicular to the channel (cross-section 2). A heavily-doped silicon wafer is used, with an electrical resistivity between 0.01 and 0.02
Ω · cm. The fabricated silicon electrodes will have the same electrical resistivity.

Figure 2a–e show the microchannel formation process. First, in Figure 2a, two rows of slits are etched into the hard mask layers using reactive ion etching (RIE). Each slit is 5 μm long and 2 μm wide. The distance between the slits is 3 μm. If the spacing, *S*, between the two rows of slits is properly designed, as shown in Figure 2b, semi-isotropic sulfur hexafluoride (SF_6_) plasma etching through the slit openings can produce two separate cavities. There is a certain amount of silicon remaining between the two microchannels. In Figure 2c, silicon dioxide (SiO_2_) is deposited using LPCVD. In Figure 2d, at the backside of the wafer, RIE is used to etch the inlet and outlet holes into the hard mask layers. Then a Bosch-based process is used to etch the inlets and outlets into the substrate until the SiO_2_ layer is reached. The temporary SiO_2_ layer is used to protect the slits pattern in the SiRN membrane during the Bosch-based process. In Figure 2e, the SiO_2_ protection layer is removed by hydrogen fluoride (HF) and SiRN is deposited using LPCVD to seal the slits and form the channel walls.

Figure 2f–h show the metallization process. Firstly, in Figure 2f, the SiRN layer at the wafer grounding locations (cross-section 1) and between two microchannels (cross-section 2) is etched by RIE. Secondly, in Figure 2g, a 5 nm thick platinum layer is sputtered and annealed at 400 ${}^{\circ }C$ to form a thin conductive layer of platinum silicide. This is followed by sputtering 10 nm tantalum, 20 nm thick platinum, and 200 nm thick gold consecutively without breaking the vacuum. Finally, in Figure 2h, the metal thin films are patterned using ion beam etching (IBE).

Figure 2i shows the channel release process. At the frontside of the wafer, the SiRN layer is etched using RIE to form release windows of width *x* next to the microchannels. Then, the channels are released using isotropic SF_6_ RIE through these windows. By properly designing the width, *W*, of the flat SiRN membrane above the microchannels, some silicon will remain between the sidewalls of the microchannels after the release etch, forming the sidewall electrode. The release recipe used is given in Table 1.

### 2.2. Test Structure Designs

Test structures are designed and fabricated using the modified SCT process to perform a parametric study on the dimensions of *S* and *W*. The design criteria for the test structures and the resulting geometry and dimensions of the silicon electrodes are presented in this section.


In the modified SCT process, the silicon electrode is supposed to remain between the sidewalls of two adjacent microchannels after the release etch. For the release etch, we chose to use a fixed release window width x= 200 μm and the same frontside release recipes as used in the SCT process in a silicon wafer [18,19]. At least two microchannels are needed to produce one silicon sidewall electrode. Three microchannels are needed to produce two silicon sidewall electrodes.

In the following sections, firstly, two typical microchannels (‘small channel’ and ‘big channel’) fabricated by the original SCT process will be presented (Figure 3). Their design parameters will be used as a reference for designing the test structures. Secondly, test structure designs using two microchannels with the small-small channel configuration are presented. Thirdly, test structure designs using three microchannels are presented. Three microchannels could become too wide to be fully released. Therefore, two test structures using three microchannels are designed with the ‘small-big-small’ and ‘big-small-big’ channel configurations to investigate the possibility of integrating two silicon electrodes using three microchannels.

#### 2.2.1. Typical Microchannel Dimensions

The cross-sections of a small channel and a big channel fabricated by the original SCT process are schematically illustrated in Figure 3. The top view of the slits array mask design for each microchannel is also displayed. The gap between two slits is 3 μm. Each slit has the same dimensions of 5 μm long and 2 μm wide as reported in the original SCT process [18,19].

One row of slits results in the small channel shown in Figure 3 [18,19]. Due to fabrication tolerances, dimensions vary slightly. Typically, the maximum channel width $D_{1}$ could range from 50 μm to 55 μm and the minimum flat membrane width $M_{1}$ could range from 40 μm to 45 μm [18,19].

Three parallel rows of slits with row spacing $S_{0}=$ 5 μm result in the big channel shown in Figure 3. Typically, the maximum channel width $D_{3}$ could range from 95 μm to 100 μm and the minimum flat membrane width $M_{3}$ could range from 80 μm to 85 μm [18,19].

The dimensions of the small channel and the big channel will be used as a reference when designing the test structures.

#### 2.2.2. Two Parallel Microchannels

The simplest design for the test structure consists of two microchannels with the small-small channel configuration. Figure 4a shows the design parameters *S* and *W* in the mask design and the cross-sections.

When the two small channels are close to each other their channel walls should not touch, so *S* should be larger than $D_{1}$. The width *W* determines how much silicon remains after the release etch; *W* should be larger than $2\cdot M_{1}$.

In the test structures, we used three different values for *S* (50, 55 and 60 μm) and nine different values for *W* (80, 90, 100, 120, 140, 160, 180, 200 and 250 μm).

#### 2.2.3. Three Parallel Microchannels

A test structure using three microchannels with the small-big-small channel configuration is shown in Figure 4b. The design parameters *S* and *W* are investigated in order to obtain two silicon electrodes.

When the three microchannels are close together their channel walls should not touch, so *S* should be at least $\frac{D_{1}+D_{3}}{2}=$
72.5
μm. *W* should be at least $D_{3}+2\cdot D_{1}=$ 210 μm. Please note that the big channel in the middle might not become free-hanging after the release etch.

Based on these constraints, in these test structures, *S* is increased from 72.5
μm to 80 μm in steps of 2.5
μm. *W* is increased from 210 μm to 340 μm in steps of 10 μm.

Other multichannel configurations are also possible for obtaining silicon electrodes. For example, test structures using small-small-small channels, big-small-big channels, and big-big channels. However, these will require further adjustments to the standard release recipes to produce electrically functional silicon electrodes. One example of a test structure using the big-small-big channel configuration will be further discussed in Section 2.3.2.

### 2.3. Test Structure Fabrication Results

The scanning electron microscope (SEM) photographs in Figure 5 and Figure 6 show cross-sectional views of the fabricated test structures using two microchannels and three microchannels. The red circles highlight the silicon electrodes.

#### 2.3.1. Two Microchannels

Figure 5a–d show the results for a constant W= 100 μm and only *S* is varied. In Figure 5a, S= 50 μm results in a single microchannel without silicon sidewall electrode. In Figure 5b, S= 55 μm results in a single microchannel with a silicon electrode with triangular cross-section inside the microchannel. In Figure 5c,d, as *S* increases to 60 μm, two separated microchannels are formed without silicon sidewall electrode.

Figure 5c–h show that a constant S= 60 μm always results in two separate microchannels and the cross-sectional area of the silicon sidewall electrode depends on *W*. Figure 5c,d show that W= 100 μm results in no silicon electrode. Figure 5e,f show W= 120 μm results in a silicon sidewall electrode with cross-sectional area of approximately 35 μm${}^{2}$. Figure 5g shows that W= 140 μm results in a silicon sidewall electrode with cross-sectional area of approximately 230 μm${}^{2}$. Figure 5h shows that at W= 250 μm, there is a maximum amount of silicon sidewall electrode with a cross-sectional area of about 720 μm${}^{2}$. Larger values of *W* will result in an incomplete release of the channels and the electrode will remain connected to the silicon substrate.

#### 2.3.2. Three Parallel Microchannels

Figure 6 shows two silicon sidewall electrodes in a small-big-small channel configuration with S=
72.5
μm and W= 330 μm. Figure 6a,b show that the big channel in the middle is still fixed to the substrate. The close-up images in Figure 6c,d show that the two silicon sidewall electrodes are in between the small channel and big channel and not connected to the substrate anymore. They are enclosed within the SiRN channel walls. A prolonged release etch could help to allow the big channel to become free-hanging. However, there is a trade-off between the cross-sectional area of the two silicon sidewall electrodes and the detachment of the big channel from the substrate.

Three microchannels can also be arranged in the big-small-big channel configuration as shown in Figure 7. There are two types of results. Figure 7a,b show that a single, wide microchannel is formed when S= 70 μm and W= 330 μm. Inside the channel, two silicon electrodes with triangular cross-section (highlighted in the red circles) are embedded underneath the top side of the channel. The silicon is detached from the substrate and thus can be used as an electrode. Figure 7b shows that after the release etch, the silicon underneath the channel is not completely removed.

Figure 7c,d show that three separated channels formed when S=72.5μm and W= 330 μm. Figure 7d shows that after the release, the silicon between the adjacent channels is still connected to the substrate thus can not be used as an electrode. The release etch profile near the big channel wall has affected the detachment of the embedded silicon from the substrate. In future research, prolonged release etch can be tested to solve this problem. Furthermore, in order to gain more control over the amount of remaining silicon, a release etch from the backside of the wafer can be investigated in future research.

## 3. Demonstrator Chip Design and Electrical Characterization

In order to experimentally characterize the electrical properties of the silicon electrodes fabricated by the modified SCT process, demonstrator chips were designed to use the silicon electrodes as microheaters. The goal is to experimentally achieve temperatures above 100 ${}^{\circ }C$ by Joule heating the silicon microheaters.

In the following sections, firstly, based on the results of the test structures, the design considerations for the silicon microheater are presented. Next, the demonstrator chip design is presented. Finally, experiments are carried out to characterize the resistance of the silicon microheater, the temperature coefficient of resistance (TCR, or α) of the metal temperature sensors, and the temperatures achieved during the Joule heating. For temperature measurement, metal resistors are integrated on top of the microchannels.

### 3.1. Microheater Design

The resistance of the silicon microheater *R* can be calculated by the estimation of the cross-sectional area A, length *l*, and the electrical resistivity ρ of the silicon with the relation $R=\rho \frac{l}{A}$. This estimation is limited to the situation where A is constant over the full length of the silicon microheater.

In comparison to a typical 10 μm wide and 200 nm thick platinum microheater, Figure 5 shows that the cross-sectional area of one silicon electrode can vary from approximately 35 μm${}^{2}$ (Figure 5e) to approximately 720 μm${}^{2}$ (Figure 5h). Table 2 lists the calculated properties of these microheaters for Joule heating, assuming an electrical resistivity of $10^{-2}$
Ω·cm for silicon and $10^{-5}$
Ω·cm for platinum [39], although a somewhat higher resistivity is expected for the platinum thin films [39]. When used as microheaters silicon electrodes provide a clear advantage, since supplying the same current density of $5\cdot 10^{11}$
$A\;m^{-2}$ through the electrodes, a factor of 10^4^ to 10^6^ more power per unit length ($\dot{P}$) can be dissipated in the silicon electrodes. This is due to the combination of higher resistance per unit length ($\dot{R}$) and larger cross-sectional areas (A) of the silicon electrodes.

For the demonstrator chips, we chose to use the design parameters for the silicon electrode shown in Figure 5h, i.e., two rows of slits with row spacing S= 60 μm, SiRN membrane width W= 250 μm, and release window width x= 200 μm. In this way, silicon microheaters with the maximum values for *A* and $\dot{P}$ can be expected. This design also allows a prolonged over-etching time during the release process to obtain the silicon electrodes shown in Figure 5e,g. Therefore, silicon microheaters with $\dot{R}$ ranging from $10^{5}$
$\Omega \;m^{-1}$ to $3\cdot 10^{6}$
$\Omega \;m^{-1}$ can be expected. When a current density of $5\cdot 10^{11}$
$A\;m^{-2}$ is supplied during Joule heating, a power, $\dot{P}$, ranging from $9\cdot 10^{8}$
$W\;m^{-1}$ to $1\cdot 10^{10}$
$W\;m^{-1}$ is dissipated in the silicon heaters.

### 3.2. Demonstrator Chip Design

A microfluidic demonstrator chip is designed to experimentally verify the electrical properties of the integrated sidewall silicon microheaters. Metal thin film temperature sensors are placed on top of the microchannels to monitor the temperature during the Joule heating of the silicon microheater.

The mask design of the 7.3
mm × 7.3
mm microfluidic demonstrator chip is shown in Figure 8. Please note that this chip design allows fluid to flow into the chip; however, no fluid flow was applied during the experiments for electrical characterization of the silicon microheaters.

Fluidic inlet and outlet holes of 300 μm × 40 μm are located at the backside of the wafer. They are connected with two frontside inlet channels labeled as Inlet 1 and Inlet 2. At the frontside of the wafer, microchannels of different hydraulic diameters are formed by choosing proper spacing between slit rows. The two inlet channels merge into a single meandering channel formed by 3 rows of slits with 10 μm row spacing, which can serve as mixing channel if needed.

Then, the mixing channel gradually transforms into two parallel swissroll-shaped microchannels where one silicon microheater is embedded in between the channels. Figure 8a shows one end of the microchannels and the “Start” of the silicon microheater, which are anchored to the substrate, and Figure 8d shows the other end of the microchannels and the “End” of the silicon microheater, which are free-hanging in the air. The suspended microchannels allow thermal expansion during Joule heating. Note that the “Start” of the microheater in Figure 8a will be electrically connected to the silicon substrate.

The integrated silicon microheater is between (Figure 8a) the “Start” and (Figure 8b) the “End” point. The red colored 10 μm × 30 μm rectangular openings indicate the ohmic contact pad connecting the metal layer to the silicon microheater. Joule heating of the silicon microheater is carried out by power supply between these two contacts. Metal temperature sensors labeled in green color are placed on top of the swissroll-shaped microchannels.

In the suspended swissroll area, the two parallel microchannels are designed to allow counter flow. Close-up image (a) shows that the fluid mixture flows into one channel until reaching the center of the swissroll area, and then flows out in the other channel. Eventually, the flow leaves the chip via the outlet channel and the outlet hole at the backside of the chip.

The swissroll-shaped microchannels are surrounded by the release cavities indicated by the white areas in Figure 8. This can greatly reduce conductive heat loss from the silicon microheater through the silicon to the substrate. The swissroll design allows the highest temperature in the chip center. Moreover, Figure 8c shows that a suspended SiRN membrane is designed in between the channel structures to maintain a constant frontside release window width of x= 200 μm next to each microchannel. Small bridges are used to connect the microchannel and the suspension membrane for mechanical strength.

Figure 9 shows optical microscope images of one of the demonstrator chips fabricated in the modified SCT process. Please note that this specific chip is not used for the electrical characterization in the next section. Figure 9a–d show close-up photographs which correspond to the design details in Figure 8a–d, respectively.

### 3.3. Electrical Characterization

For electrical characterization, a demonstrator chip is mounted on printed circuit board and electrical connections are made by wire bonding as shown in Figure 10a. A close-up photograph of the chip is shown in Figure 10b.

Figure 10c shows a schematic drawing of the chip where the swissroll shape is represented by straight channels. One end of the microchannels is anchored on the substrate and the silicon microheater is connected to the substrate, while the other end is free-hanging in the air. The silicon microheater is located between the two parallel small channels (purple color). The ohmic contacts for the silicon microheaters are labeled as “Start” and “End”. Two thin film metal resistors on top of the structure are used as temperature sensors (TS). TS1 (blue color) is located near the free-hanging end of the channels, and TS2 (red color) is located approximately halfway in the channels.

In the following sections, we present the experiments and results for the demonstrator chip to characterize the performance of the silicon electrode as a microheater.

#### 3.3.1. Electrical Resistance

A four-wire resistance measurement was used to measure the electrical resistance of the silicon microheater using a Keithley 2602 system source meter. The measured resistance of the silicon microheater is *R* = 15,870 Ω. The designed silicon microheater is *l* = 15,560 μm long. This corresponds to a resistance per unit length $\dot{R}=\frac{R}{l}\approx 10^{6}$
$\Omega \;m^{-1}$. The extrinsic silicon wafer has a resistivity ρ ranging from 0.01 Ω
cm to 0.02 Ω
cm. The cross-sectional area *A* of the fabricated silicon microheater in the demonstrator chip can be calculated by $A=\frac{\rho \cdot l}{R}$ which ranges from 98 μm${}^{2}$ to 196 μm${}^{2}$. According to the cross-sectional areas of the test structures, as shown in Table 2, the amount of silicon remaining in the demonstrator chip is in between the cross-sectional views shown in Figure 5e,g. Although attempts were made to make photographs of the channel cross-sections by SEM, the suspended swissroll shape proved to be too fragile for this.

During the demonstrator chip fabrication run, the release etch was not uniform. Some demonstrator chips had discontinuous silicon microheaters, which was verified by the measured resistances. Therefore, it is possible that the cross-sectional area of the silicon microheater shown in Figure 10 is also not uniform. In future work, development of a more uniform release etch recipe would help to achieve reproducible resistance per unit length in the silicon microheater. A thorough investigation on the wafer-scale non-uniformity and proximity effect during the SF_6_ release etch shall be continued.

#### 3.3.2. Temperature Coefficient of Resistance

As mentioned earlier, two thin film metal temperature sensors located on top of the microchannels are used to monitor the temperature *T* of the microchannel.

The TCR or α of the metal temperature sensors can be experimentally characterized following the relation [40]:(1)$$R=R_{ref}+\alpha \cdot R_{ref}\cdot \left(T-T_{ref}\right)$$
which can be rewritten as:(2)$$R=\alpha \cdot R_{ref}\cdot T+R_{ref}\cdot \left(1-\alpha \cdot T_{ref}\right)$$

A high temperature compatible TCR measurement setup was customized for the TCR characterization experiment. The demonstrator chip was placed inside a Heraeus T5025 oven, customized with electrical readout and controlled by a National Instruments LabVIEW program. The oven temperature was controlled to increasefrom 45 ${}^{\circ }C$ to 80 ${}^{\circ }C$ in steps of 5 ${}^{\circ }C$. Each temperature step was stabilized for 20 min and monitored with a thermocouple inside the oven. At each stabilized temperature step, 300 data points of the resistance *R* are consecutively measured within 5 min. Figure 11 shows the averaged *R* at each stable temperature step *T* for the two temperature sensors. A linear fit is also shown for both data sets.

Table 3 shows the reference resistance $R_{ref}$ at the reference oven temperature $T_{ref}$ for each temperature sensor. Equation (Equation 2) and the intercept with the y-axis $R_{ref}\cdot \left(1-\alpha \cdot T_{ref}\right)$ from Figure 11 are used to calculate the TCR value. For the multilayer thin film temperature sensor a TCR of $1.22\cdot 10^{-3}$
${}^{\circ }C^{-1}$ is obtained, which is smaller than the TCR value for either bulk gold ($3.4\cdot 10^{-3}$
${}^{\circ }C^{-1}$) [41] or bulk platinum ($3.9\cdot 10^{-3}$
${}^{\circ }C^{-1}$) [41]. For each pure metal thin film, the thin film deposition process introduces strain, grain boundaries, and defects compared to their bulk crystalline counterparts [41]. Therefore, typically higher resistivities and lower TCR values are expected in thin films. Furthermore, the substrate temperature increases during the consecutive sputtering of the multilayered metal thin films. Thin film platinum and gold form an alloy at 200 ${}^{\circ }C$ in a very short time [42]. The obtained TCR values agree with the experimental literature which reported that the measured TCR value for a platinum/gold alloy was about $\frac{1}{3}$ of pure platinum or pure gold [41].

#### 3.3.3. Joule Heating of the Silicon Microheater

In order to experimentally verify the capability of using the silicon electrodes as microheaters, a Joule heating experiment of the silicon microheaters was carried out.

A voltage *V* was supplied to the silicon microheater using a Keithley power source. The current *I* through the silicon microheater was simultaneously measured by a Keithley k2000 multimeter. The electric power *P* dissipated in the silicon microheater is then given by P=IV.

During the Joule heating, the resistances *R* of TS1 and TS2 were measured by a Keithley 2602 system source meter. The temperature *T* of the two temperature sensors can be calculated by rewriting Equation (Equation 1) into:(3)$$T=T_{ref}+\frac{\frac{R}{R_{ref}}-1}{\alpha }$$

Figure 12 shows the measured temperatures of TS1 and TS2 as a function of the dissipated power. A maximum heating power of 206.9
mW was generated in the silicon microheaters during the experiment. TS1 monitored a highest temperature of 119.4
${}^{\circ }C$ and TS2 monitored a highest temperature of 80.5
${}^{\circ }C$.

These temperatures are within the expected range when the heat loss to the environment is taken into consideration. There is no internal flow in the microchannels and temperature elevation is relatively low, therefore heat convection and radiation can be neglected. This means that conduction through the channel material and conduction through the surrounding air will be the dominant heat loss mechanisms. As the channel is more than 15 mm long and the distance from the substrate is less than 200 μm we assume that conduction through the air will be dominant. An order of magnitude for the expected temperature elevation can be easily calculated by assuming a semi-circular geometry as indicated in Figure 13. Assuming a uniform temperature $T_{channel}$ for the channel and a uniform temperature $T_{substrate}$ for the silicon substrate, and taking conduction through the air as the sole heat loss mechanism, the temperature difference is given by [43]:(4)$$T_{channel}-T_{substrate}=\frac{q}{\pi \cdot l\cdot k_{air}}\cdot \ln\left(\frac{r_{substrate}}{r_{channel}}\right)$$

Inserting approximate values of the power q= 200 mW, the channel length l= 16 mm, the thermal conductivity for air $k_{air}=$
0.033
W
m^−1^K^−1^ (at 120 ${}^{\circ }C$), the radius of the air cavity $r_{substrate}=$ 200 μm and the radius of the channel $r_{channel}=$ 60 μm results in a temperature difference of 145 ${}^{\circ }C$. This is higher than the measured temperature elevation of approximately 100 ${}^{\circ }C$, but the difference can be easily explained by conduction through the channel itself and convection.

Moreover, conduction through the air also explains that a higher temperature was measured by TS1 than by TS2, because TS1 is located in the center of the swissroll-shaped chip and is surrounded by hot microchannels. TS2 is located at the edge of the chip and only one side is next to hot channels. This also confirms that the design of a swissroll-shaped chip allows the highest temperature in the chip center. In future research, the heat loss by air conduction can be minimized by performing the Joule heating experiment in vacuum.

## 4. Conclusions and Outlook

Optimization of the mask designs in the Surface Channel Technology allows to embed bulk silicon electrodes between the sidewalls of the adjacent free-hanging microfluidic channels. The fabricated silicon electrodes have cross-sectional areas ranging from approximately 35 μm${}^{2}$ to 720 μm${}^{2}$ and a resistance per unit length ranging from approximately $1\cdot 10^{5}$
Ω m^−1^ to $3\cdot 10^{6}$
Ω m^−1^.

A demonstrator chip employing the silicon electrode as a microheater embedded between the adjacent free-hanging microchannels is designed and fabricated. A power of 206.9
mW is dissipated in the silicon microheater by Joule heating and the thin film metal temperature sensor measures 119.4
${}^{\circ }C$. This proves that the embedded silicon is isolated from the conductive substrate so that it can be used as an electrode. Calculations show that the measured temperature elevation is limited by conduction through the surrounding air. Significantly higher temperatures could be achieved in a vacuum environment.

With the optimized mask design, all the currently existing microfluidic devices fabricated using the original SCT process can be integrated with silicon electrodes. This is one step closer to reach the goal of multi-parameter devices.

In future research, improvement of the release etch recipe would help to achieve a more uniform silicon microheater resistance as is designed. To demonstrate the robustness as a power resistor compared to the thin film metal microheater, Joule heating experiments shall be conducted in vacuum to reach higher temperatures. Furthermore, investigations shall be continued to explore the possibilities and advantages of integrating the silicon sidewall electrode in various microfluidic applications.

## Figures and Tables

**Figure 1 micromachines-11-00561-f001:**
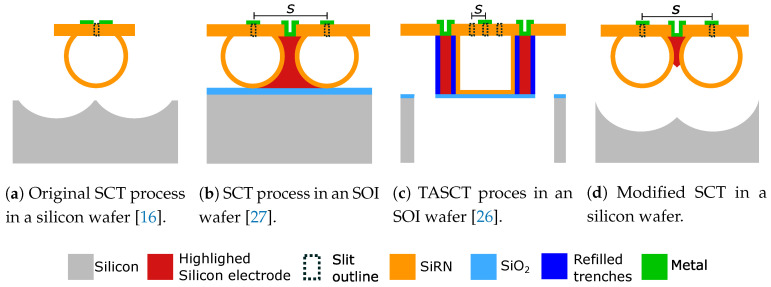
Schematic cross-sectional illustrations of four types of microchannels fabricated by four variations of the Surface Channel Technology (SCT) process. A microchannel made by (**a**) the original SCT process in a silicon wafer uses one row of slits (dashed lines) and uses thin film metal electrodes on top of the silicon–rich silicon nitride (SiRN) membranes for sensing or actuation functions. Microchannels made by (**b**) SCT process in an SOI wafer, (**c**) Trench-Assisted Surface Channel Technology (TASCT) process in an SOI wafer, and (**d**) modified SCT process in a silicon wafer all allow to embed the silicon electrodes (highlighted in red) onto the sidewalls of the microchannels. The dimension and location of the microchannels in (**b**,**d**) are determined by the spacing *S* between two rows of slits, and in (**c**) by the locations of the refilled trenches.

**Figure 2 micromachines-11-00561-f002:**
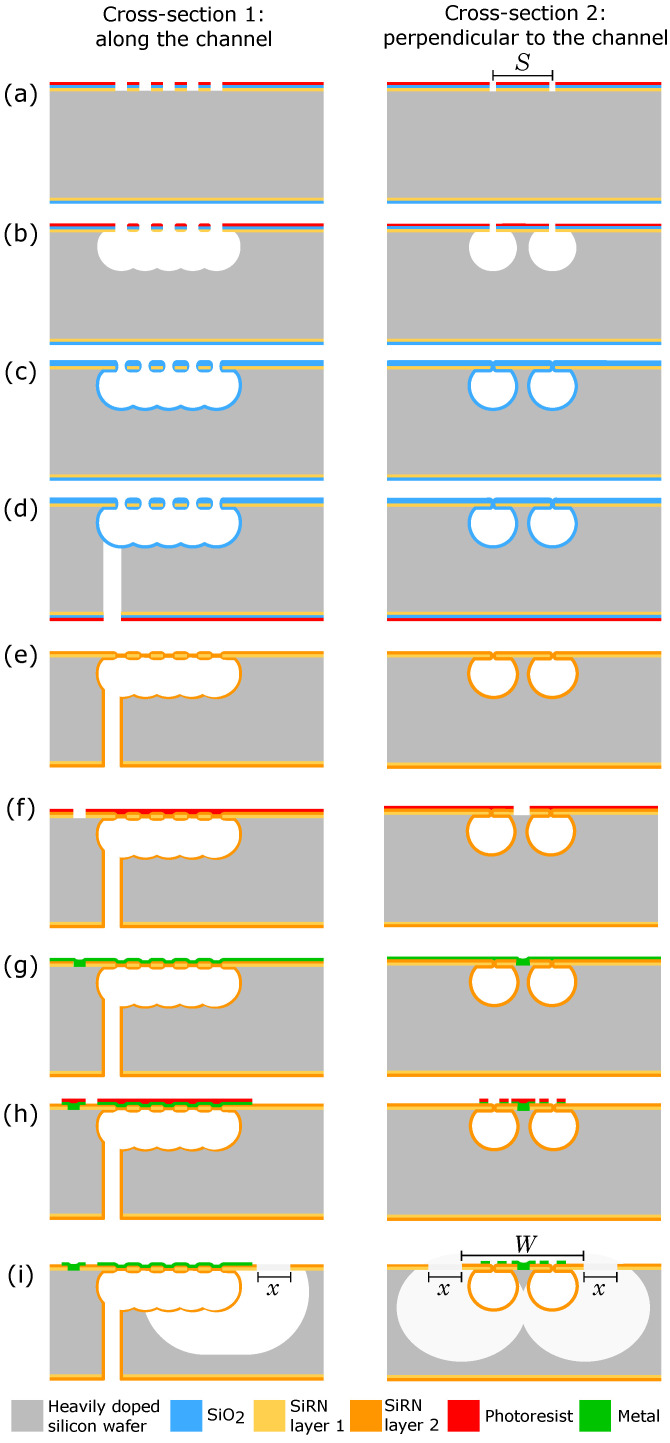
The modified SCT fabrication process is schematically illustrated in cross-sectional views along the channel length (cross-section 1) and perpendicular to the channel (cross-section 2). (**a**) Start with a heavily doped silicon wafer. RIE is used to etch SiO_2_ and SiRN to form two rows of slits with spacing *S*; (**b**) semi-isotropic RIE the silicon through the slits. If *S* is designed properly, two separate microchannels will be formed; (**c**) deposit SiO_2_ using LPCVD; (**d**) RIE is used to etch SiO_2_ and SiRN at the backside of the wafer. Bosch-based DRIE of the inlet and outlet holes until reaching SiO_2_; (**e**) wet etching of SiO_2_ using HF. Seal the slits and form SiRN channel walls using LPCVD. (**f**) Etch SiRN above the silicon sidewall electrode (cross-section 2) and the device grounding (cross-section 1) using RIE; (**g**) sputtering of and (**h**) IBE the metal thin films for electrical interconnections; (**i**) etching of the release windows of width *x* by directional RIE and releasing the channels by isotropic RIE. If the width *W* of the flat SiRN membrane above the microchannels is designed properly, a silicon sidewall electrode will remain between the microchannels.

**Figure 3 micromachines-11-00561-f003:**
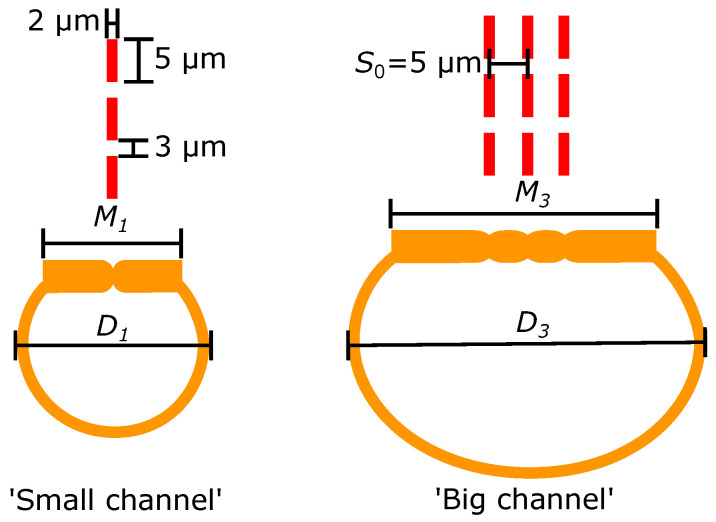
Cross-sectional illustrations of two microchannels fabricated by the original SCT process [18,19]. The small channel uses one row of slits and the big channel uses three rows of slits. The slit rows are indicated by the red dashed lines.

**Figure 4 micromachines-11-00561-f004:**
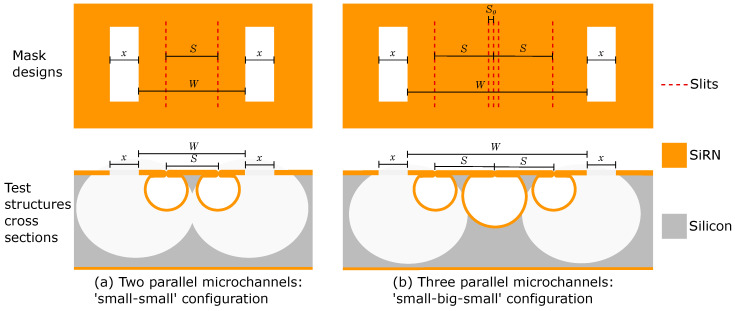
Mask designs and cross-sectional views of the test structures using (**a**) two small channels and (**b**) three channels with the small-big-small configuration. Design parameters *S* and *W* are varied in the test structures.

**Figure 5 micromachines-11-00561-f005:**
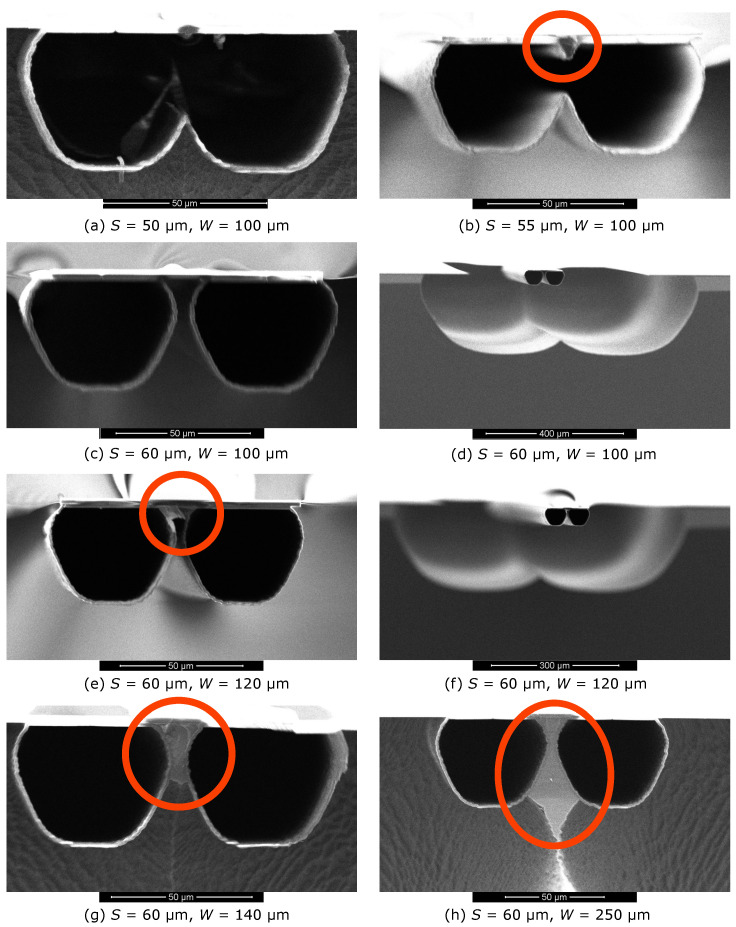
SEM photographs show the cross-sectional views of the fabricated test structures using two microchannels with the ’small-small’ channel configuration. The produced silicon electrode is highlighted in the red circle.

**Figure 6 micromachines-11-00561-f006:**
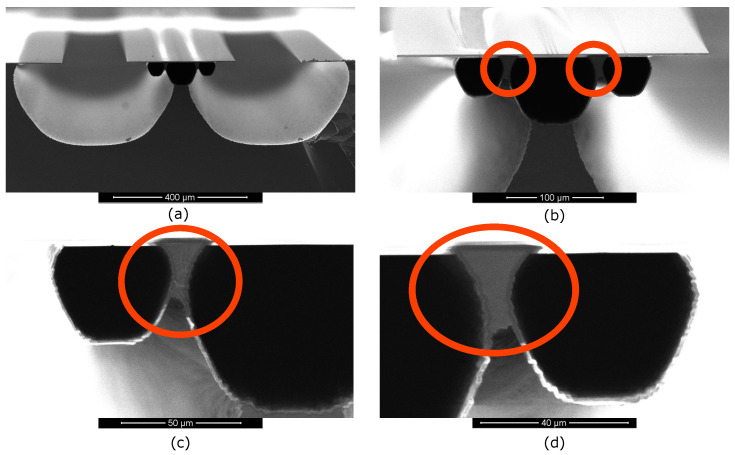
SEM photographs showing the cross-sectional views of fabricated test structures using three microchannels with the small-big-small channel configuration. The produced silicon electrodes are highlighted in the red circles.

**Figure 7 micromachines-11-00561-f007:**
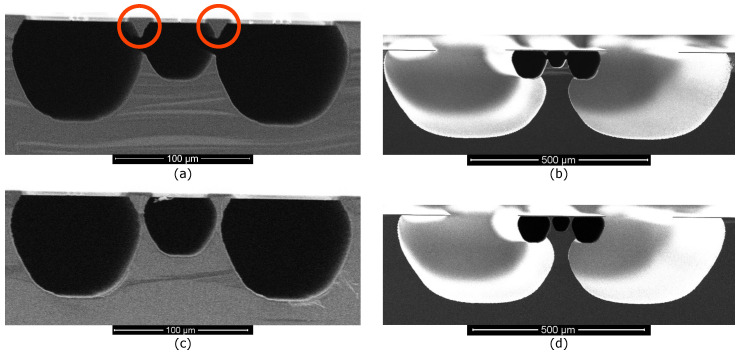
SEM photographs show the cross-sectional views of the fabricated test structures using three microchannels with the big-small-big channel configuration. In (**a**,**b**) two silicon electrodes with triangular cross-sections are highlighted in the red circles. In (**c**,**d**) no silicon electrodes are produced.

**Figure 8 micromachines-11-00561-f008:**
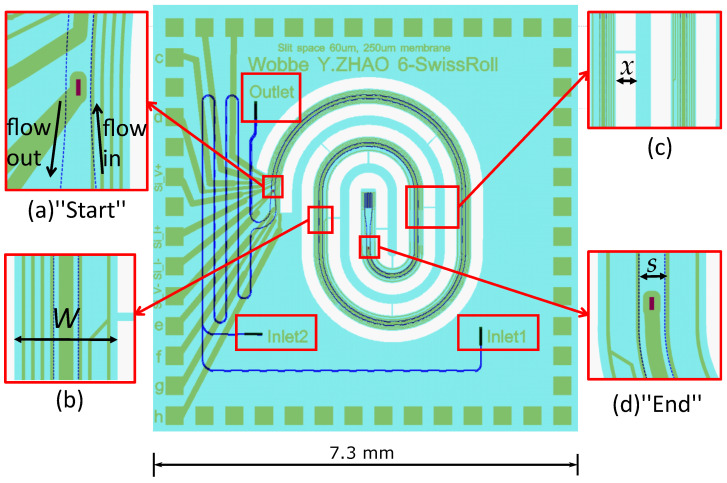
The demonstrator chip mask designs show that the swissroll-shaped microchannels and the embedded microheater are suspended. Metal wires (green color) are placed on top of the SiRN microchannels (blue color). The close-up views in (**a**,**d**) show two electrical contacts for the silicon microheater. Two parallel rows of slits form two separated microchannels and one silicon microheater is embedded in between. The close-up view in (**b**) shows the SiRN membrane width of W= 250 μm, in (**c**) shows the release windows of width of x= 200 μm, and in (**d**) shows the row spacing of S= 60 μm.

**Figure 9 micromachines-11-00561-f009:**
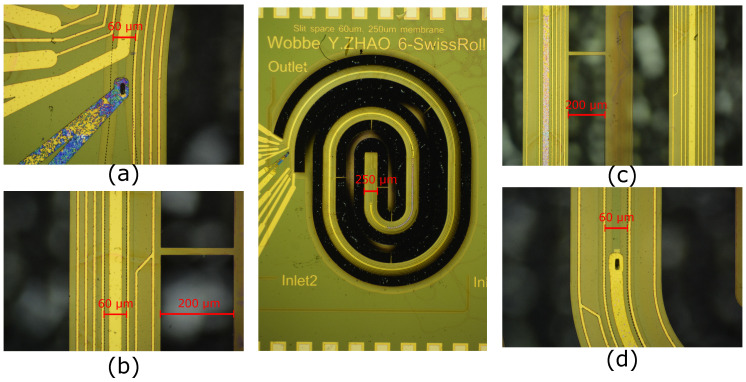
The optical microscope images of a demonstrator chip fabricated by the modified SCT process.

**Figure 10 micromachines-11-00561-f010:**
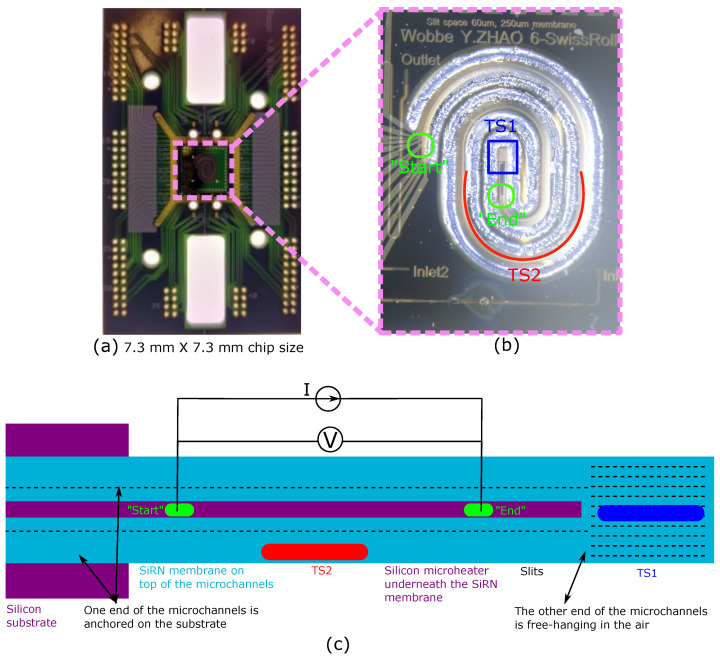
Photographs of (**a**) a demonstrator chip mounted on a printed circuit board and (**b**) a close-up of the swissroll-shaped channels in the demonstrator chip. A schematic drawing is shown in (**c**), representing the swissroll-shaped channels by a straight structure.

**Figure 11 micromachines-11-00561-f011:**
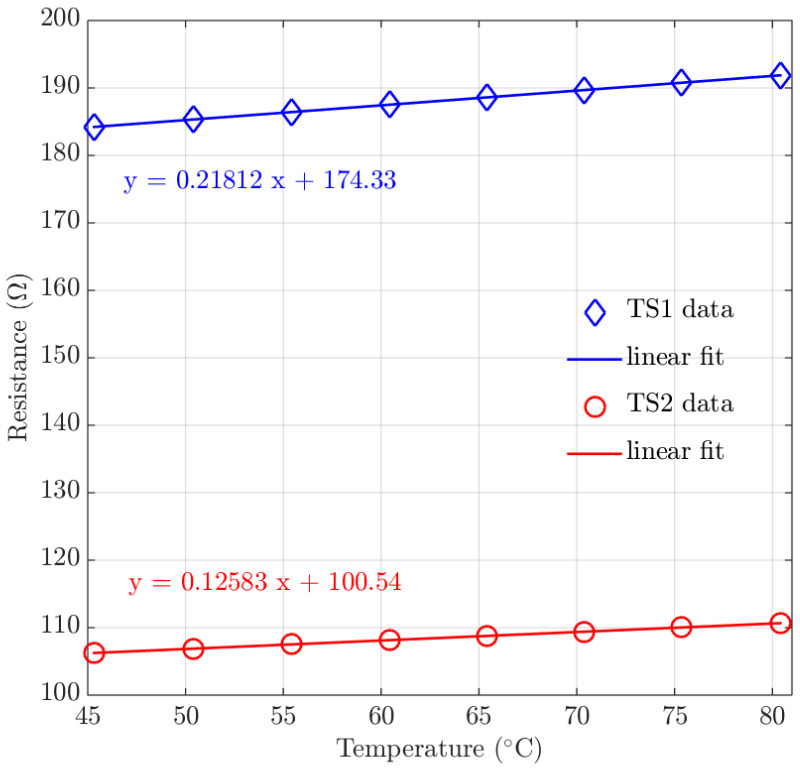
Plot of the measured resistance versus the oven temperature. The intercept with the y-axis is used to calculate the TCR of temperature senors.

**Figure 12 micromachines-11-00561-f012:**
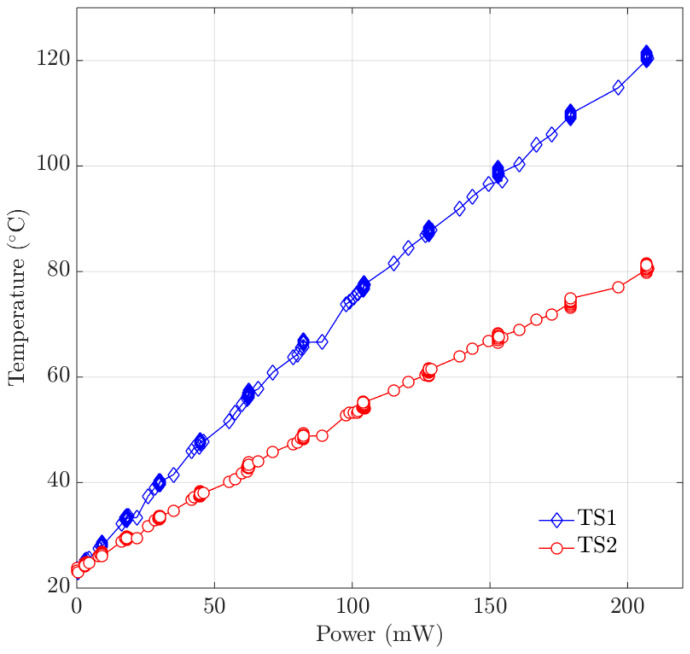
Joule heating result of the silicon microheater in the demonstrator chip.

**Figure 13 micromachines-11-00561-f013:**
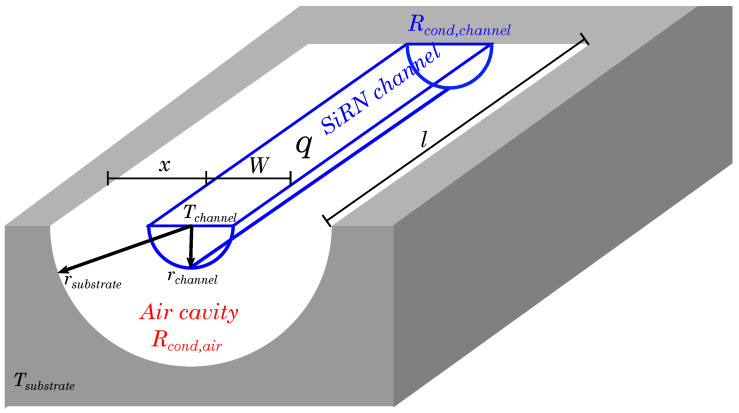
A simplified heat conduction model using a semi-circular air cavity underneath a suspended semi-circular microchannel.

**Table 1 micromachines-11-00561-t001:** Settings of the isotropic reactive ion etching (RIE) release etch recipe on the SPTS Pegasus and Oxford Instruments PlasmaPro 100 Estrelas.

Parameter	Recipe
Platen Temperature	20 ${}^{\circ }C$
SF_6_ flow	600 sccm
Pressure	90 mTorr/8% valve
ICP	3000 W
CCP	Off
Etch time	7 × 5 min (with 2 min interval)

**Table 2 micromachines-11-00561-t002:** Comparison between thin film platinum and bulk silicon as microheaters for Joule heating. At a fixed current density of $5\cdot 10^{11}$$A\;m^{-2}$, silicon microheaters can dissipate much more power per unit length than the platinum microheaters.

Microheaters	Resistivity (ρ)	Cross-Sectional Area (*A*)	Resistance per Unit Length ($\dot{R}$)	Power Generation per Unit Length ($\dot{P}$) (Supply a Current Density of $5\cdot 10^{11}$ $A\;m^{-2}$)
Thin film platinum	$10^{-5}$ Ω·cm	2 μm${}^{2}$	$5\cdot 10^{4}$ $\Omega \;m^{-1}$	$5\cdot 10^{4}$ $W\;m^{-1}$
Silicon (Figure 5e)	$10^{-2}$ Ω·cm	35 μm${}^{2}$	$3\cdot 10^{6}$ $\Omega \;m^{-1}$	$9\cdot 10^{8}$ $W\;m^{-1}$
Silicon (Figure 5g)	$10^{-2}$ Ω·cm	230 μm${}^{2}$	$4\cdot 10^{5}$ $\Omega \;m^{-1}$	$5\cdot 10^{9}$ $W\;m^{-1}$
Silicon (Figure 5h)	$10^{-2}$ Ω·cm	720 μm${}^{2}$	$1\cdot 10^{5}$ $\Omega \;m^{-1}$	$1\cdot 10^{10}$ $W\;m^{-1}$

**Table 3 micromachines-11-00561-t003:** TCR calculations based on Equation (Equation 2) and the intercept with the y-axis in Figure 11.

Temperature Sensor	$T_{\mathrm{ref}}$	$R_{\mathrm{ref}}$	$R_{\mathrm{ref}}\cdot \left(1-\alpha \cdot T_{\mathrm{ref}}\right)$	α
TS1	25.08 ${}^{\circ }C$	179.83 Ω	174.33	$1.22\cdot 10^{-3}$ ${}^{\circ }C^{-1}$
TS2	25.08 ${}^{\circ }C$	103.71 Ω	100.54	$1.22\cdot 10^{-3}$ ${}^{\circ }C^{-1}$

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
