# Peer review of "Heavily-Doped Bulk Silicon Sidewall Electrodes Embedded between Free-Hanging Microfluidic Channels by Modified Surface Channel Technology"

_micromachines, 2020, doi:10.3390/mi11060561_

Round 1

Reviewer 1 Report

The manuscript presents a new technology to integrate efficient heaters in suspended microchannels. The approach is clever, there are several potential applications, the presentation is generally clear and convincing, the experimental part is well designed, and the conclusions are supported by the results.

My only general criticism is that the process, unlike the one in [9], looks critically dependent on the control of two SF6 etches, and especially the one that defines the silicon heaters. The manuscript suggests that this has been an issue: see lines 177-179, the large difference between the expected (720 um­­2 for device of fig. 5h from table 1) and actual (line 266, not declared but probably around five to ten times smaller, see also point 4. below) cross-section, and the paragraph at lines 260-273. The authors should be more explicit on this problem.

Specifically, I think certain minor corrections are required for publication, a few of which are related to the above criticism:

  1. Section 1.1 and 1.2 do not include a thorough presentation of the literature. Of the 14 references, I counted 11 from the authors. The justification given for this work is essentially internal, i.e. it is an evolution of their previous work, but suspended channels have been and are actively investigated by other groups: you need to present the current status of research on this subject and compare your previous and current work with the literature.
  2. Lines 86-89: does the amount of silicon left after release depend on W alone, or also (and substantially) by the SF6 etch parameters and etching time? If, as I suspect, the latter is true, you should state it clearly here.
  3. In fig. 8, a quoted line marking the dimension of chip side would be nice, as would be dimension markers in the different photos of fig. 9.
  4. lines 269-273: please state explicitly the estimated cross-section area (not just “between the test structures shown in Figure 5e and Figure 5g”), based on your resistance measurements. If you have any data from additional samples other that the ones of Table 1 (as is suggested by the fact that you had to exclude a few with discontinuous resistors), include them in the table, or give a quantitative estimate of the cross-section area deviation, if this is possible at all. Also, even if you pursue the “development of uniform release etch recipes” in the future, you cannot completely avoid wafer disuniformities (e.g. both differences between wafer center and border, and wafer-to-wafer variation), and you should mention this intrinsic limitations. Please also state if all your samples come from the same zone of a single wafer or not, because in the former case etch disuniformity might be much worse that the one you observed.
  5. Lines 280-281: A few more details on the oven setup would be useful. You should at least state how the sample temperature T was measured, how you determined that the temperature was stable at each temperature point, and how long you had to wait between each temperature point.
  6. The TCR value in table 2 is much smaller than the one you would expect from a gold/platinum resistor (alpha_Au ~ 0.003, alpha_Pt ~ 0.004). Do you have any hypotheses for this discrepancy? Also, on line 288, can you explain what is meant by “as-deposited”?
  7. Line 323: after having developed a simple analytical model for DeltaT, you do not give its value computed from Eq. (6), but only the “order of magnitude.” Even if the number is far from the measured one (but within an order of magnitude), you should state its value. A sensible reader should not expect that such a simple model would give an accurate prediction: it was developed just to check that your experimental data were not unreasonable. If you feel that the model is too crude and the number too bad, you should then remove the modeling part altogether. After all, this is a fabrication/characterization paper, not a modeling paper.

Finally, there is a typo on line 82 (“braking” instead of “breaking”).

Author Response

Reply to reviewer 1

The authors would like to thank the reviewer for the detailed comments. These are highly appreciated.

The following replies are based on the content in version SCT_micromachines - manuscript - marked, where newly added sentences are marked in blue, and deleted sentences are marked with strikethrough lines. Another version “SCT_micromachines - manuscript - final” without the editing traces is also uploaded.

The manuscript presents a new technology to integrate efficient heaters in suspended microchannels. The approach is clever, there are several potential applications, the presentation is generally clear and convincing, the experimental part is well designed, and the conclusions are supported by the results.

My only general criticism is that the process, unlike the one in [9], looks critically dependent on the control of two SF6 etches, and especially the one that defines the silicon heaters. The manuscript suggests that this has been an issue: see lines 177-179, the large difference between the expected (720 um2 for device of fig. 5h from table 1) and actual (line 266, not declared but probably around five to ten times smaller, see also point 4. below) cross-section, and the paragraph at lines 260-273. The authors should be more explicit on this problem.

The homogeneity of the release etch has indeed been a limitation. We have emphasized this more in the manuscript, both in the main text, lines 330-334, and in the conclusion.

Specifically, I think certain minor corrections are required for publication, a few of which are related to the above criticism:

  1. Section 1.1 and 1.2 do not include a thorough presentation ofthe literature. Of the 14 references, I counted 11 from the authors. The justification given for this work is essentially internal, i.e. it is an evolution of their previous work, but suspended channels have been and are actively investigated by other groups: you need to present the current status of research on this subject and compare your previous and current work with the literature.

Reply: Sections 1.1 and 1.2 have been extended significantly and now give an overview of the various types of microchannels that have been proposed in literature and of the electrodes that are used. We also better explain the limitations of using thin-film metal electrodes. A large number of references was added.

  1. Lines 86-89: does the amount of silicon left after release depend on Walone, or also (and substantially) by the SF6 etch parameters and etching time? If, as I suspect, the latter is true, you should state it clearly here.

Reply: Yes, the amount of silicon indeed depends on both the design parameter W and the SF6 release etch parameters and etching time. We kept the release etch recipe the same as in the original SCT process, but for completeness we have added table 1 with the recipe that was used. We also emphasize more that a longer release etch reduces the cross-sectional area of the electrodes (lines 214-215).

  1. In fig. 8, a quoted line marking the dimension of chip side would be nice, as would be dimension markers in the different photos of fig. 9.

Reply: This was indeed not clear. In Fig. 8 we have now indicated the dimensions of the chip of 7.3 mm. In Fig. 9 the dimensions of features are added to show the scale.

  1. lines 269-273: please state explicitly the estimated cross-section area (not just “between the test structures shown in Figure 5e and Figure 5g”), based on your resistance measurements. If you have any data from additional samples other that the ones of Table 1 (as is suggested by the fact that you had to exclude a few with discontinuous resistors), include them in the table, or give a quantitative estimate of the cross-section area deviation, if this is possible at all. Also, even if you pursue the “development of uniform release etch recipes” in the future, you cannot completely avoid wafer disuniformities (e.g. both differences between wafer center and border, and wafer-to-wafer variation), and you should mention this intrinsic limitations. Please also state if all your samples come from the same zone of a single wafer or not, because in the former case etch disuniformity might be much worse that the one you observed.

Reply: We used the resistivity of the silicon wafer to obtain an estimate for the average cross-sectional area and we now added this calculation in lines 320-326. In lines 330-334 we emphasize that the release etch was not uniform and needs to be improved. The demonstrator chips were fabricated in a multi-project fabrication run together with other chips. Therefore, unfortunately we have only a few devices and no conclusions can be drawn with respect to the nonuniformity over the wafer. Most other devices were “standard SCT devices” without the silicon electrodes. We did examine the released cavities by SEM and found that there are dirt or impurities in the cavities which we normally do not see. The same release recipe was also used for the test structures, with much better homogeneity.

We definitely agree that there is an intrinsic non-uniformity, which is caused by both the wafer-scale and the local loading effects from the release windows. We have added a new sentence in line 334-336, stating that “A thorough investigation on the wafer-scale non-uniformity and proximity effect during the SF6 release etch shall be continued.”.

  1. Lines 280-281: A few more details on the oven setup would be useful. You should at least state how the sample temperature Twas measured, how you determined that the temperature was stable at each temperature point, and how long you had to wait between each temperature point.

Reply: More details are added in lines 342-350. Each temperature step was stabilized for 20 minutes and monitored with a thermocouple inside the oven. At each stabilized temperature step, 300 data points of the resistance R are consecutively measured within 5 minutes.

  1. The TCR value in table 2 is much smaller than the one you would expect from a gold/platinum resistor (alpha_Au ~ 0.003, alpha_Pt ~ 0.004). Do you have any hypotheses for this discrepancy? Also, on line 288, can you explain what is meant by “as-deposited”?

Reply: In lines 355-362 we added a discussion to explain this discrepancy. By “as-deposited” thin films, we meant that the sputtered films do not receive any post-processing such as high temperature annealing. We have rewritten the sentence.

  1. Line 323: after having developed a simple analytical model for DeltaT, you do not give its value computed from Eq. (6), but only the “order of magnitude.” Even if the number is far from the measured one (but within an order of magnitude), you should state its value. A sensible reader should not expect that such a simple model would give an accurate prediction: it was developed just to check that your experimental data were not unreasonable. If you feel that the model is too crude and the number too bad, you should then remove the modeling part altogether. After all, this is a fabrication/characterization paper, not a modeling paper.

Reply: The calculation was indeed only meant to check that the measured values are not unreasonable. We have now shortened the derivation and added the resulting value of 145 oC. This is a bit higher than measured, but seems reasonable given that the estimate is based on only the conduction through air and ignores conduction through the channel itself.

Finally, there is a typo on line 82 (“braking” instead of “breaking”).

Reply: Thank you for noticing this. We changed it into “breaking”.

Reviewer 2 Report

Review Comment

This paper acknowledges its originality in that it introduces a new microfluidic chip fabrication method and a micro-heater using this method. Despite this, however, it was unfavorable in terms of the aim and practicality that the manuscript sought. This paper simply listed what they did, and suggested no solutions to the researchers. The following is a dictation of the problems I am thinking of, so please refer to them.

  1. Abstract

 (1) The first sentence of the abstract mentions platinum or gold as thin film metal electrodes, and suddenly silicon electrodes appeared. Of course, I understand the author's intentions, but I think it would be a good idea to explain this background very well in abstract or put it in the introduction part.

 (2) It is not well understood whether microfluidic fabrication method or micro-heater is important. If both are important, please rewrite the two correlations properly in the sentence.

  à I think it would be nice to claim that this manuscript invents new microfluidic chip fabrication processes and use this method to fabricate a micro-heater to achieve these good results.

  1. Introduction

 (1) I found some errors in grammar, so please check again.

 (2) Figure 1 makes it easy to understand the originality of this paper.

 (3) Page 2 Line 36: Compared to Reference 9 and your manuscript, you claim that this method doesn’t need an electrical insulation. What are the advantages of this research experimentally revealed compared to reference 9? This is because the microfluidic fabrication processes of this paper are not simple and practical, so this method may not be used if there is no advantage compared to the reference 9

(4) It would be necessary to remove 1.1, 1.2 and 1.3, merge them into one, and change only the paragraph.

  1. Modified surface channel technology

(1) Modified surface channel technology is recommended to be changed to Experiment.

(2) The two views in Figure 2 are confusing to understand microfluidic chip fabrication. It seems that presenting cross-section 2 may be helpful for understanding.

(3) I wonder why the test structure design was done. Is it simply a release problem in chip fabrication? If so, you should devise a way to reduce tension when releasing. I simply think that one channel, two channels, and three channels are not desirable. I think it will help to protect the structure if Teflon coating (a few microns, less than 10µm thickness) is done after RIE process.

  1. Demonstrator chip design and electrical characterization

 (1) I recommend that changing from “Demonstrator chip design and electrical characterization” to “Results and discussion”

 (2) Page 9, Line 197: Use reference.

 (3) It is recommended to combine Figures 8 and 9 into one.

 (4) Page 11, Line 247: Why do you use PCB?

 (5) Page 12, Line 267: This manuscript mention that suspended swissroll-shaped channels are easy to fragile. Without any resolution, it seems that this method is not useful to use it.

 (6) Page 12, Line 278: It is said that Eq.1 was obtained experimentally, but it is not understood. Could it be derived from Figure 11?

 (7) Page 13, Line 297: How do you get this equation? If you use any reference, you have to mention it?

  1. Conclusions

 (1) Conclusions need some clarification on what the results were, but two paragraphs (line 331, line 335) briefly mention chip fabrication and micro-heater. It is necessary to mention what experiment results were obtained from the micro-heater through a new method of chip fabrication, and I think that simply listing what you have done is a meaningless sentence.

 (2) Future research comes up with a thought that needs to be mentioned. The problems in this study are presented, but it is not a good manuscript to simply list them without suggesting solutions.

Author Response

Reply to reviewer 2

The authors would like to thank the reviewer for the detailed comments. These are highly appreciated.

The following replies are based on the content in version SCT_micromachines - manuscript - marked, where newly added sentences are marked in blue, and deleted sentences are marked with strikethrough lines. Another version “SCT_micromachines - manuscript - final” without the editing traces is also uploaded.

This paper acknowledges its originality in that it introduces a new microfluidic chip fabrication method and a micro-heater using this method. Despite this, however, it was unfavorable in terms of the aim and practicality that the manuscript sought. This paper simply listed what they did, and suggested no solutions to the researchers. The following is a dictation of the problems I am thinking of, so please refer to them.

  1. Abstract
  • The first sentence of the abstract mentions platinum or gold as thin film metal electrodes, and suddenly silicon electrodes appeared. Of course, I understand the author's intentions, but I think it would be a good idea to explain this background very well in abstract or put it in the introduction part.

Reply: We have rewritten the abstract (lines 5-10) and now emphasize the limitations of thin-film metal electrodes and that we present a method to realize silicon sidewall electrodes that do not have these limitations.

 (2) It is not well understood whether microfluidic fabrication method or micro-heater is important. If both are important, please rewrite the two correlations properly in the sentence.

I think it would be nice to claim that this manuscript invents new microfluidic chip fabrication processes and use this method to fabricate a micro-heater to achieve these good results.

Reply: Thank you for the suggestion. We have rewritten the abstract in line 7-10 as follows: “In this paper, we report an innovative idea to make microfluidic devices with integrated silicon sidewall electrodes, and we demonstrate their use as microheaters. This is achieved by modifying the original Surface Channel Technology with optimized mask designs.”

  1. Introduction
  • I found some errors in grammar, so please check again.

Reply: We thoroughly checked the entire manuscript for grammar and spelling mistakes and we fixed several mistakes.

  • Figure 1 makes it easy to understand the originality of this paper.

Reply: Thank you very much.

 (3) Page 2 Line 36: Compared to Reference 9 and your manuscript, you claim that this method doesn’t need an electrical insulation. What are the advantages of this research experimentally revealed compared to reference 9? This is because the microfluidic fabrication processes of this paper are not simple and practical, so this method may not be used if there is no advantage compared to the reference 9.

Reply: We extended and improved the introduction, adding more references, so reference [9] has now become [26], see figure 1(b). These are the two main reasons why our work is not replaceable by reference [26]:

  • Firstly, [26] requires an SOI wafer to realize the silicon electrodes. When possible a standard silicon wafer is preferred.
  • In [26] there is no easy way to release the structure from the substrate. To do this, one would need to etch a cavity from the backside of the wafer.
  • It would be necessary to remove 1.1, 1.2 and 1.3, merge them into one, and change only the paragraph.

Reply: We do not understand why the reviewer requests this change. We kept the subsections, but extended them to provide a better comparison with existing literature.

  1. Modified surface channel technology
  • Modified surface channel technology is recommended to be changed to Experiment.

Reply: We also thought about using the section title from the LaTeX template, but we prefer to use a descriptive title for each section. This section is about the fabrication technology and also gives the results for the test structures that were designed and realized.

  • The two views in Figure 2 are confusing to understand microfluidic chip fabrication. It seems that presenting cross-section 2 may be helpful for understanding.

Reply: We think both cross-sectional views are essential to fully understand the process and the fabricated devices. In figure 2 above the figure, we added the description “Cross-section 1: along the channel” and “Cross-section 2: perpendicular to the channel".  

  • I wonder why the test structure design was done. Is it simply a release problem in chip fabrication? If so, you should devise a way to reduce tension when releasing. I simply think that one channel, two channels, and three channels are not desirable. I think it will help to protect the structure if Teflon coating (a few microns, less than 10µm thickness) is done after RIE process.

Reply: At the beginning of section 2.2, lines 138-140, the purpose of the test structures is now clarified: “Test structures are designed and fabricated using the modified SCT process to perform a parametric study on the dimensions S and W. The design criteria for the test structures and the resulting geometry and dimensions of the silicon electrode are presented in this section.”  
The results are used in the next section to design the demonstrator chip.

  1. Demonstrator chip design and electrical characterization
  • I recommend that changing from “Demonstrator chip design and electrical characterization” to “Results and discussion”

Reply: We also thought about using the template section name, but decided that it was not suitable.

  • Page 9, Line 197: Use reference.

Reply: Reference [39] is added. A new sentence “although a somewhat higher resistivity is expected for the platinum thin films [39]” is added in line 246-247.

  • It is recommended to combine Figures 8 and 9 into one.

Reply: Thank you for the recommendation. We have thought about this, but we think that a combined figure would contain too much information. Therefore, we decide to keep separate figures, but we do emphasize that the close-up photographs in figure 9 correspond exactly to the clos-up pictures in figure 8.

  • Page 11, Line 247: Why do you use PCB?

Reply: This is now explained in lines 301-303 “For electrical characterization, a demonstrator chip is mounted on printed circuit board and electrical connections are made by wire bonding”.

  • Page 12, Line 267: This manuscript mention that suspended swissroll-shaped channels are easy to fragile. Without any resolution, it seems that this method is not useful to use it.

Reply: The sentence is rewritten in line 328-329. We state that the swissroll-shaped free-hanging microchannels are fragile in the sense that they cannot be broken at an exact location (for making a photograph) by using tweezers. There is no problem in using the structure for its intended application.

  • Page 12, Line 278: It is said that Eq.1 was obtained experimentally, but it is not understood. Could it be derived from Figure 11?

Reply: Thank you for pointing this out. Equation [1] is commonly used for describing the temperature dependence of resistors and we did not find it experimentally. We have now added reference [40] in line 341.

  • Page 13, Line 297: How do you get this equation? If you use any reference, you have to mention it?

Reply: Reference [40] is added to Eq.1 and it is now emphasized in the manuscript that Eq. 2 and Eq.3 are obtained by rewriting Eq.1.

  1. Conclusions
  • Conclusions need some clarification on what the results were, but two paragraphs (line 331, line 335) briefly mention chip fabrication and micro-heater. It is necessary to mention what experiment results were obtained from the micro-heater through a new method of chip fabrication, and I think that simply listing what you have done is a meaningless sentence.

Author reply: The conclusions have been rewritten, see the three paragraphs in line 409-422.

  • Future research comes up with a thought that needs to be mentioned. The problems in this study are presented, but it is not a good manuscript to simply list them without suggesting solutions.

Author reply: This paragraph has also been rewritten, see lines 423-427.

Round 2

Reviewer 2 Report

No further comment